# L2B: Learning to Bootstrap for Combating Label Noise

## Abstract

Deep neural networks are powerful tools for representation learning, but can easily overfit to noisy labels which are prevalent in many real-world scenarios. Generally, noisy supervision could stem from variation among labelers, label corruption by adversaries, etc. To combat such label noises, one popular line of approach is to apply customized weights to the training instances, so that the corrupted examples contribute less to the model learning. However, such learning mechanisms potentially erase important information about the data distribution and therefore yield suboptimal results. To leverage useful information from the corrupted instances, an alternative is the bootstrapping loss, which reconstructs new training targets on-the-fly by incorporating the network's own predictions (i.e, pseudo-labels).

In this paper, we propose a more generic learnable loss objective which enables a joint reweighting of instances and labels at once. Specifically, our method dynamically adjusts the *per-sample importance weight* between the real observed labels and pseudo-labels, where the weights are efficiently determined in a meta process. Compared to the previous instance reweighting methods, our approach concurrently conducts implicit relabeling, and thereby yield substantial improvements with almost no extra cost. Extensive experimental results demonstrated the strengths of our approach over existing methods on multiple natural and medical image benchmark datasets, including CIFAR-10, CIFAR-100, ISIC2019 and Clothing 1M. Code will be made publicly available.

## 1 Introduction

Recent advances in deep learning have achieved great success on various computer vision applications, where large-scale clean datasets are available. However, noisy labels or intentional label corruption by an adversarial rival could easily cause dramatic performance drop (Nettleton et al., 2010). This problem is even more crucial in the medical field, given that the annotation quality requires great expertise. Therefore, understanding, modeling, and learning with noisy labels has gained great momentum in recent research efforts (Frénay & Verleysen, 2013; Natarajan et al., 2013; Han et al., 2019; Li et al., 2019; Liu et al., 2020; Jiang et al., 2018; Ren et al., 2018; Xue et al., 2019; Li et al., 2020; Wang et al., 2020; Zheng et al., 2021; Yao et al., 2021; Zhu et al., 2021; Wu et al., 2021; Zhou et al., 2021).

Existing methods of learning with noisy labels primarily take a loss correction strategy. One popular direction is to first estimate the noise corruption matrix and then use it to correct the loss function (Patrini et al., 2017; Goldberger & Ben-Reuven, 2017). However, correctly estimating the noise corruption matrix is usually challenging and often involves assumptions about the noise generation process (Xia et al., 2019; Liu & Tao, 2015; Hendrycks et al., 2018). Other research efforts focus on selecting clean samples from the noisy data (Jiang et al., 2018; Han et al., 2018; Yu et al., 2019; Fang et al., 2020) by treating samples with small loss as clean ones (Arpit et al., 2017). Instead of directly discarding those "unclean" examples, an extension of this idea is focusing on assigning learnable weights to each example in the noisy training set (Ren et al., 2018; Shu et al., 2019), where noisy samples have low weights. However, discarding or attending less to a subset of the training data (e.g., noisy samples) can erase important information about the data distribution.

To fully exploit the corrupted training samples, another direction is to leverage the network predictions (i.e., pseudo-labels (Lee et al., 2013)) to correct or reweight the original labels (Reed et al.,

2014; Tanaka et al., 2018), so that the holistic data distribution information could be preserved during network training. One representative work is the bootstrapping loss (Reed et al., 2014), which introduces a perceptual consistency term in the learning objective that assigns a weight to the pseudo-labels to compensate for the erroneous guiding of noisy samples. While in this strategy, the weight for the pseudo-labels is manually selected and remains the same for all training samples, which does not prevent fitting the noisy ones and can even lead to low-quality label correction (Arazo et al., 2019). To tackle this challenge, Arazo et al. (Arazo et al., 2019) designed a dynamic bootstrapping strategy to adjusts the label weight by fitting a mixture model. Instead of separately reweighting labels or instances, in this paper, we propose a more generic learning strategy to enable a joint instance and label reweighting. We term our method as **L**earning to **B**ootstrap (**L2B**), where we aim to leverage the learner's own predictions to bootstrap itself up for combating label noise from a meta-learning perspective.

During each training iteration, L2B learns to dynamically re-balance the importance between the real observed labels and pseudo-labels, where the per-sample weights are determined by the validation performance on a separated clean set in a meta network. Unlike the bootstrapping loss used in (Reed et al., 2014; Arazo et al., 2019; Zhang et al., 2020) which explicitly conducts relabeling by taking a weighted sum of the pseudo- and the real label, L2B reweights the two losses associated with the pseudo- and the real label instead (where the weights need not be summed as 1). In addition, we theoretically prove that our formulation, which reweights different loss terms, can be reduced to the original bootstrapping loss and therefore conducts an implicit relabeling instead. By learning these weights in a meta-process, our L2B yields substantial improvement (e.g., **+8.9%** improvement on CIFAR-100 with 50% noise) compared with the instance reweighting baseline with almost no extra cost. We conduct extensive experiments on public natural image datasets (i.e., CIFAR-10, CIFAR-100, and Clothing 1M) and medical image dataset (i.e., ISIC2019), under different types of simulated noise and real-world noise. Our method outperforms various existing explicit label correction and instance reweighting works, demonstrating the strengths of our approach.

Our main contributions are as follows:

- We propose a generic learnable loss objective which enables a joint instance and label reweighting, for combating label noise in deep learning models.

- We prove that our new objective is, in fact, a more general form of the bootstrapping loss, and propose L2B to efficiently solve for the weights in a meta-learning framework.

- Compared with previous instance re-weighting methods, L2B exploits noisy examples more effectively without discarding them by jointly re-balancing the contribution of real and pseudo labels.

- We show the theoretical convergence guarantees for L2B, and superior results on natural and medical image recognition tasks under both synthetic and real-world noise.

## 2 RELATED WORKS

**Learning through explicit relabeling.** To effectively handle noisy supervision, many works propose to directly correct the training labels through estimating the noise transition matrix (Xia et al., 2019; Yao et al., 2020; Goldberger & Ben-Reuven, 2017; Patrini et al., 2017) or modeling noise by graph models or neural networks (Xiao et al., 2015; Vahdat, 2017; Veit et al., 2017; Lee et al., 2018). Patrini et al. (Patrini et al., 2017) estimate the label corruption matrix to directly correct the loss function. Hendrycks et al. (Hendrycks et al., 2018) further propose to improve the corruption matrix by using a clean set of data, which then enables training a corrected classifier. However, these methods usually require assumptions about noise modeling. For instance, Hendrycks et al. (Hendrycks et al., 2018) assume that the noisy label is only dependent on the true label and independent of the data. Another line of approaches proposes to leverage the network prediction for explicit relabeling. Some methods (Tanaka et al., 2018; Yi & Wu, 2019) relabel the samples by directly using pseudo-labels in an iterative manner. Han et al. use generated prototypes as pseudo-labels to be more noise tolerant (Han et al., 2019). Instead of assigning the pseudo-labels as supervision, Reed et al. (Reed et al., 2014) propose to generate new training targets by a convex combination of the real and pseudo labels. In a recent study, Ortego et al. (Ortego et al., 2021) directly apply this strategy for classification refinement, and combine it with contrastive learning for training noise-robust models. However,

using a fixed weight for all samples does not prevent fitting the noisy ones could even limit the label correction. To tackle this challenge, Arazo et al. propose a dynamic bootstrapping strategy, which calculates sample weights by modeling per-sample loss with a beta mixture model (Arazo et al., 2019). Zhang et al. further propose to learn this weight in a meta step, and combine semi-supervised learning for furthering the performance (Zhang et al., 2020).

**Instance reweighting.** To reduce the negative effect of corrupted examples, many research efforts have also been dedicated to selecting or reweighting training instances so that noisy samples contribute less to the loss (Jiang et al., 2018; Ren et al., 2018; Fang et al., 2020). Based on the observation deep neural networks tend to learn simple patterns first before fitting label noise (Arpit et al., 2017), many methods treat samples with small loss as clean ones (Jiang et al., 2018; Shen & Sanghavi, 2019; Han et al., 2018; Yu et al., 2019; Wei et al., 2020). Among those methods, Co-teaching (Han et al., 2018) and Co-teaching+ (Yu et al., 2019) train two networks where each network selects small-loss samples in a mini-batch to train the other. Li et al. (Li et al., 2020) further propose to incorporate semi-supervised learning techniques to better leverage the noisy examples. Jiang et al. (Jiang et al., 2018) propose to use curriculum learning to improve convergence and generalization by ordering instances. Rather than directly selecting clean examples for training, meta-learning-based instance reweighting methods are also gaining momentum recently (Ren et al., 2018; Shu et al., 2019; Xu et al., 2021). In these methods, the example weights and the network parameters are updated in a bi-level optimization to determine the contribution of each training sample. This line of approach has also been successfully applied for robust medical image analysis (Xue et al., 2019; Mirikharaji et al., 2019).

Different from the aforementioned approaches which separately handle instance reweighting and label reweighting, we propose a new general learning objective to simultaneously adjust the per-sample loss weight while implicitly relabeling the training samples.

## 3 METHODOLOGY

### 3.1 PRELIMINARY

Given a set of $N$ training samples, i.e., $\mathcal{D}_{tra} = \{(x_i, y_i) | i = 1, ..., N\}$, where $x_i \in \mathbb{R}^{W \times H}$ denotes the $i$-th image and $y_i$ is the observed noisy label. In this work, we also assume that there is a small unbiased and clean validation set $\mathcal{D}_{val} = \{(x_i^v, y_i^v) | i = 1, ..., M\}$ and $M \ll N$, where the superscript $v$ denotes the validation set. Let $\mathcal{F}(:, \theta)$ denote the neural network model parameterized by $\theta$. Given an input-target pair $(x, y)$, we consider the loss function of $\mathcal{L}(\mathcal{F}(x, \theta), y)$ (e.g., cross-entropy loss) to minimize during the training process. Our goal, in this paper, is to properly utilize the small validation set $\mathcal{D}_{val}$ to guide the model training on $\mathcal{D}_{tra}$, for reducing the negative effects brought by the noisy annotation.

To establish a more robust training procedure, Reed et al. proposed the bootstrapping loss (Reed et al., 2014) to enable the learner to "disagree" with the original training label, and effectively relabel the data during the training. Specifically, the training targets will be generated using a convex combination of training labels and predictions of the current model (i.e., pseudo-labels (Lee et al., 2013)), for purifying the training labels. Therefore, for a $L$-class classification problem, the loss function for optimizing $\theta$ can be derived as follows:

$$y_i^{\text{pseudo}} = \arg\max_{l=1,..,L} \mathcal{P}(x_i, \theta), \tag{1}$$

$$\theta^* = \arg\min_{\theta} \sum_{i=1}^{N} \mathcal{L}(\mathcal{F}(x_i, \theta), \beta y_i^{\text{real}} + (1 - \beta) y_i^{\text{pseudo}}), \tag{2}$$

where $\beta$ is used for balancing the weight between the real labels and the pseudo-labels. $\mathcal{P}(x_i, \theta)$ is the model output. $y^{\text{real}}$ and $y^{\text{pseudo}}$ denote the observed label and the pseudo-label respectively. However, in this method, $\beta$ is manually selected and fixed for all training samples, which does not prevent fitting the noisy ones and can even lead to low-quality label correction (Arazo et al., 2019). Moreover, we observe that this method is quite sensitive to the selection of the hyper-parameter $\beta$. For instance, as shown in Figure 1(a), even a similar $\beta$ selection (i.e., $\beta = 0.6/\beta = 0.8$) behaves differently under disparate noise levels, making the selection of $\beta$ even more intractable. Another

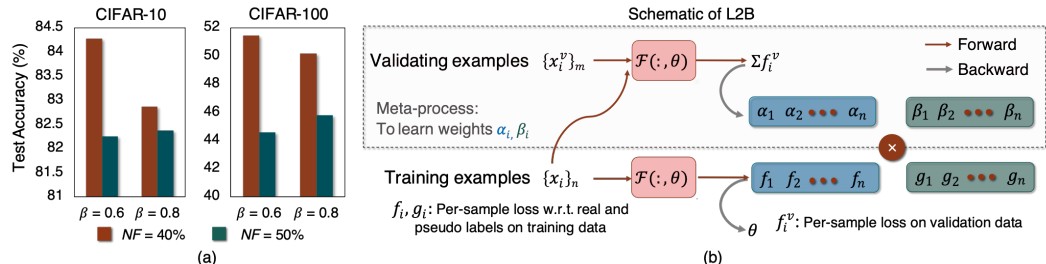

Figure 1: (a) The original bootstrapping loss (Reed et al., 2014) is sensitive to the reweighting hyper-parameter $\beta$. Under different noise levels, the optimal $\beta$ is different (*NF* stands for noise fraction). (b) Schematic description of our Learning to Bootstrap (i.e., **L2B**) method. The reweighting hyper-parameters are learned in a meta-process.

limitation lies in that Eq. equation 2 treats all examples as equally important during training, which could easily cause overfitting for biased training data.

### 3.2 Learning to Bootstrap through Meta-Learning

To address these above challenges, in this paper, we aim to learn to bootstrap the model by conducting a joint label reweighting and instance reweighting. To achieve this, we propose to generate meta-learned weights for guiding our main learning objective:

$$\theta^*(\boldsymbol{\alpha}, \boldsymbol{\beta}) = \arg\min_{\theta} \sum_{i=1}^{N} \alpha_i \mathcal{L}(\mathcal{F}(x_i, \theta), y_i^{\text{real}})$$
$$+ \beta_i \mathcal{L}(\mathcal{F}(x_i, \theta), y_i^{\text{pseudo}}), \tag{3}$$

with $\{\alpha_i, \beta_i\}_{i=1}^N$ being the balance weights. Here we note that this new learning objective can be regarded as a general form of the original bootstrapping loss, as Eq. equation 3 can be reduced to Eq. equation 2 when $\alpha_i + \beta_i = 1$ given that $\mathcal{L}(\cdot)$ is the cross-entropy loss (see details in Appendix B.1). By relaxing this constraint such that $\boldsymbol{\alpha}, \boldsymbol{\beta} \geq \mathbf{0}$, we can see that the optimization of Eq. equation 3 not only allows the main learner to explore the optimal combination between the two loss terms but also concurrently adjust the contribution of different training samples. In addition, compared with Eq. equation 2, the optimization of Eq. equation 3 does not rely on explicitly generating new training targets (i.e., $\beta y_i^{\text{real}} + (1 - \beta) y_i^{\text{pseudo}}$), but rather conducts implicit relabeling during training by reweighting different loss terms. We note that the key to L2B is that the sum of $\alpha_i$ and $\beta_i$ need not be 1, which results in **+8.9%** improvement on CIFAR-100 with 50% noise (Section 4.2).

Note that this form is also similar to self-distillation in (Li et al., 2017). But different from (Li et al., 2017) where the weights are determined by heuristics, our weights $\boldsymbol{\alpha}, \boldsymbol{\beta}$ are meta-learned based on its performance on the validation set $\mathcal{D}_{val}$, that is

$$\boldsymbol{\alpha}^*, \boldsymbol{\beta}^* = \arg\min_{\boldsymbol{\alpha}, \boldsymbol{\beta} \geq \mathbf{0}} \frac{1}{M} \sum_{i=1}^{M} \mathcal{L}(\mathcal{F}(x_i^v, \theta^*(\boldsymbol{\alpha}, \boldsymbol{\beta})), y_i^v). \tag{4}$$

It is necessary to constrain $\alpha_i, \beta_i \geq 0$ for all $i$ to avoid potential unstable training (Ren et al., 2018). Both the meta learner (i.e., Eq. equation 4) and the main learner (i.e., Eq. equation 3) are optimized concurrently, which allows the model to maximize the performance on the clean validation set $\mathcal{D}_{val}$ by adjusting the importance weights of the observed and the pseudo-labels in a differentiable manner.

**Online Approximation.** For each step $t$ at training, a mini-batch of training examples $\{(x_i, y_i), 1 \leq i \leq n\}$ with $n \ll N$ is sampled to estimate a temporary adjustment to the parameters based on the descent direction of the loss function. For simplicity, let $f_i(\theta)$ denote $\mathcal{L}(\mathcal{F}(x_i, \theta), y_i^{\text{real}})$ and $g_i(\theta)$ denote $\mathcal{L}(\mathcal{F}(x_i, \theta), y_i^{\text{pseudo}})$ in the following sections. Given any $\boldsymbol{\alpha}, \boldsymbol{\beta}$, we use

$$\hat{\theta}_{t+1} = \theta_t - \lambda \nabla \left( \sum_{i=1}^{n} \alpha_i \, f_i(\theta) + \beta_i \, g_i(\theta) \right) \Big|_{\theta=\theta_t} \tag{5}$$

---

**Algorithm 1** Learning to Bootstrap

---

**Require:** $\theta_0, \mathcal{D}_{tra}, \mathcal{D}_{val}, n, m, L$
**Ensure:** $\theta_T$
1: **for** $t = 0 \dots T - 1$ **do**
2: $\quad \{x_i, y_i\} \leftarrow \text{SampleMiniBatch}(\mathcal{D}_{tra}, n)$
3: $\quad \{x_i^v, y_i^v\} \leftarrow \text{SampleMiniBatch}(\mathcal{D}_{val}, m)$
4: $\quad$ For the $i$-th sample of $\mathcal{D}_{tra}$, compute $y_i^{\text{pseudo}} = \arg\max_{l=1,\dots,L} \mathcal{P}(x_i, \theta_t)$
5: $\quad$ Learnable weights $\boldsymbol{\alpha}, \boldsymbol{\beta}$
6: $\quad$ Compute training loss $l_f \leftarrow \sum_{i=1}^n \alpha_i f_i(\theta_t) + \beta_i g_i(\theta_t)$
7: $\quad \hat{\theta}_{t+1} \leftarrow \theta_t - \lambda \nabla l_f \big|_{\theta=\theta_t}$
8: $\quad$ Compute validation loss $l_g \leftarrow \frac{1}{m} \sum_{i=1}^m f_i^v(\hat{\theta}_{t+1})$
9: $\quad (\boldsymbol{\alpha}_t, \boldsymbol{\beta}_t) \leftarrow -\eta \nabla l_g \big|_{\boldsymbol{\alpha}=\mathbf{0}, \boldsymbol{\beta}=\mathbf{0}}$
10: $\quad \tilde{\alpha}_{t,i} \leftarrow \max(\alpha_{t,i}, 0), \ \tilde{\beta}_{t,i} \leftarrow \max(\beta_{t,i}, 0)$
11: $\quad \tilde{\alpha}_{t,i} \leftarrow \frac{\tilde{\alpha}_{t,i}}{\sum_i^n \tilde{\alpha}_{t,i} + \tilde{\beta}_{t,i}}, \ \tilde{\beta}_{t,i} \leftarrow \frac{\tilde{\beta}_{t,i}}{\sum_i^n \tilde{\alpha}_{t,i} + \tilde{\beta}_{t,i}}$
12: $\quad$ Apply learned weights $\boldsymbol{\alpha}, \boldsymbol{\beta}$ to reweight the training loss as $\hat{l}_f \leftarrow \sum_{i=1}^n \alpha_{t,i} f_i(\theta_t) + \beta_{t,i} g_i(\theta_t)$
13: $\quad \theta_{t+1} \leftarrow \theta_t - \lambda \nabla \hat{l}_f \big|_{\theta=\theta_t}$
14: **end for**

---

to approach the solution of Eq. equation 3. Here $\lambda$ is the step size. We then estimate the corresponding optimal $\boldsymbol{\alpha}, \boldsymbol{\beta}$ as

$$\boldsymbol{\alpha}_t^*, \boldsymbol{\beta}_t^* = \arg\min_{\boldsymbol{\alpha}, \boldsymbol{\beta} \geq \mathbf{0}} \frac{1}{M} \sum_{i=1}^M f_i^v(\hat{\theta}_{t+1}). \tag{6}$$

However, directly solving for Eq. equation 6 at every training step requires too much computation cost. To reduce the computational complexity, we apply one step gradient descent of $\boldsymbol{\alpha}_t, \boldsymbol{\beta}_t$ on a mini-batch of validation set $\{(x_i^v, y_i^v), 1 \leq i \leq m\}$ with $m \leq M$ as an approximation. Specifically,

$$(\alpha_{t,i}, \beta_{t,i}) = -\eta \nabla \big(\sum_{i=1}^m f_i^v(\hat{\theta}_{t+1})\big)\Big|_{\alpha_i=0, \beta_i=0}, \tag{7}$$

where $\eta$ is the step size for updating $\boldsymbol{\alpha}, \boldsymbol{\beta}$. To ensure that the weights are non-negative, we apply the following rectified function:

$$\tilde{\alpha}_{t,i} = \max(\alpha_{t,i}, 0), \ \tilde{\beta}_{t,i} = \max(\beta_{t,i}, 0). \tag{8}$$

To stabilize the training process, we also normalize the weights in a single training batch so that they sum up to one:

$$\tilde{\alpha}_{t,i} = \frac{\tilde{\alpha}_{t,i}}{\sum_i^n \tilde{\alpha}_{t,i} + \tilde{\beta}_{t,i}}, \ \tilde{\beta}_{t,i} = \frac{\tilde{\beta}_{t,i}}{\sum_i^n \tilde{\alpha}_{t,i} + \tilde{\beta}_{t,i}}. \tag{9}$$

Finally, we estimate $\theta_{t+1}$ based on the updated $\boldsymbol{\alpha}_t, \boldsymbol{\beta}_t$ so that $\theta_{t+1}$ can consider the meta information included in $\boldsymbol{\alpha}_t, \boldsymbol{\beta}_t$:

$$\theta_{t+1} = \theta_t - \lambda \nabla \big(\sum_{i=1}^n \alpha_{t,i} \, f_i(\theta) + \beta_{t,i} \, g_i(\theta)\big)\Big|_{\theta=\theta_t}. \tag{10}$$

See Appendix B.2 for detailed calculation of the gradient in Eq. equation 10. A schematic description of our Learning to Bootstrap algorithm is illustrated in Figure 1(b) and the overall optimization procedure can be found in Algorithm 1.

### 3.3 CONVERGENCE ANALYSIS

In proposing Eq. equation 3, we show that with the first-order approximation of $\boldsymbol{\alpha}, \boldsymbol{\beta}$ in Eq. equation 7 and some mild assumptions, our method guarantees to convergence to a local minimum point

of the validation loss, which yields the best combination of $\boldsymbol{\alpha}, \boldsymbol{\beta}$. Details of the proof are provided in Appendix B.3.

**Theorem 1.** *Suppose that the training loss function $f, g$ have $\sigma$-bounded gradients and the validation loss $f^v$ is Lipschitz smooth with constant L. With a small enough learning rate $\lambda$, the validation loss monotonically decreases for any training batch B, namely,*

$$G(\theta_{t+1}) \leq G(\theta_t), \tag{11}$$

*where $\theta_{t+1}$ is obtained using Eq. equation 10 and G is the validation loss*

$$G(\theta) = \frac{1}{M} \sum_{i=1}^{M} f_i^v(\theta), \tag{12}$$

*Furthermore, Eq. equation 11 holds for all possible training batches only when the gradient of validation loss function becomes 0 at some step t, namely, $G(\theta_{t+1}) = G(\theta_t) \; \forall B \Leftrightarrow \nabla G(\theta_t) = 0$*

## 4 EXPERIMENTS

### 4.1 DATASETS

**CIFAR-10 & CIFAR-100.** Both CIFAR-10 and CIFAR-100 contain 50K training images and 10K test images of size $32 \times 32$. Following previous works (Tanaka et al., 2018; Kim et al., 2019; Li et al., 2020), we experimented with both *symmetric* and *asymmetric* label noise. In our method, we used 1,000 clean images in the validation set $\mathcal{D}_{val}$ following (Jiang et al., 2018; Ren et al., 2018; Shu et al., 2019; Hendrycks et al., 2018; Zheng et al., 2021).
**ISIC2019.** Following (Xue et al., 2019), we also evaluated our algorithm on a medical image dataset, i.e., skin lesion classification data, under different symmetric noise levels. Our experiments were conducted on the 25,331 dermoscopic images of the 2019 ISIC Challenge[1], where we used 20400 images as the training set $\mathcal{D}_{tra}$, 640 images as the validation set $\mathcal{D}_{val}$, and tested on 4291 images.
**Clothing 1M.** We evaluate on real-world noisy dataset, Clothing 1M (Xiao et al., 2015), which has 1 million training images collected from online shopping websites with labels generated from surrounding texts. In addition, the Clothing 1M also provides an official validation set of 14,313 images and a test set of 10,526 images. Implementation details can be found in Appendix A.1.

### 4.2 PERFORMANCE COMPARISONS

**Experiments on CIFAR-10 & CIFAR-100.** We compare our method with different baselines: 1) Bootstrapping, which modifies the training loss by generating new training targets , 2) Distillation (Li et al., 2017), which transfers the knowledge distilled from the small clean dataset; 3) L2RW (Ren et al., 2018), which reweights different instances through meta-learning; 4) GLC (Hendrycks et al., 2018), which uses the trusted clean data to correct losses; and 5) Cross-Entropy (the standard training) under different levels of symmetric labels noise ranging from $20\% \sim 50\%$. To ensure a fair comparison, we report the best epoch for all comparison approaches. All results are summarized in Table 1 and Table 3, which shows L2B significantly outperforms all other competing methods by a large margin across all noise fractions. It is also observed that compared with previous meta-learning-based instance reweighting method L2RW, the performance improvement is substantial especially under larger noise fraction, which suggests the advantages of jointly reweighting different loss terms. For example, on CIFAR-100, the accuracy improvement of our proposed L2B reaches $7.6\%$ and $8.9\%$ under $40\%$ and $50\%$ noise fraction, respectively.

We also test our model with asymmetric noise labels (i.e., $40\%$ noise fractions) and summarize the testing accuracy in Table 2. Among all compared methods, we re-implement L2RW under the same setting and report the performance of all other competitors from previous papers (Kim et al., 2019; 2021; Li et al., 2020). Compared with previous meta-learning-based instance reweighting methods (i.e., LR2W (Ren et al., 2018), MW-Net (Shu et al., 2019)), a dynamic bootstrapping method (M-correction (Arazo et al., 2019)) and other explicit relabeling methods (i.e., F-correction (Patrini et al., 2017), Tanaka et al. (Tanaka et al., 2018)), our L2B achieves superior performance with $40\%$

---
[1]https://challenge2019.isic-archive.com/data.html

Table 1: Comparison of different methods in test accuracy (%) on CIFAR-10 with symmetric noise. *NF* stands for the noise fraction.

| Method | CIFAR-10 | | | |
|---|---|---|---|---|
| | 20% *NF* | 30% *NF* | 40% *NF* | 50% *NF* |
| Cross-Entropy | 86.9 | 84.9 | 83.3 | 81.3 |
| Bootstrapping (Reed et al., 2014) | 85.2 | 84.8 | 82.9 | 79.2 |
| Distillation (Li et al., 2017) | 88.0 | 86.8 | 85.5 | 80.0 |
| GLC (Hendrycks et al., 2018) | 91.4 | 90.3 | 88.5 | 86.4 |
| L2RW (Ren et al., 2018) | 90.6 | 89.0 | 86.6 | 85.3 |
| **L2B (Ours)** | **92.2** | **90.7** | **89.9** | **88.5** |

Table 2: Comparison with asymmetric noise.

| Method | CIFAR-10 |
|---|---|
| Cross-Entropy | 85.0 |
| F-correction (Patrini et al., 2017) | 87.2 |
| M-correction (Arazo et al., 2019) | 87.2 |
| Chen et al. (Chen et al., 2019) | 88.6 |
| P-correction (Yi & Wu, 2019) | 88.5 |
| Tanaka et al. (Tanaka et al., 2018) | 88.9 |
| MLNT (Li et al., 2019) | 89.2 |
| L2RW (Ren et al., 2018) | 89.2 |
| MW-Net (Shu et al., 2019) | 89.7 |
| NLNL (Kim et al., 2019) | 89.9 |
| JNPL (Kim et al., 2021) | 90.7 |
| **L2B (Ours)** | **91.8** |

Table 3: Comparison with different methods in test accuracy (%) on CIFAR-100 with symmetric noise. *NF* stands for the noise fraction.

| Method | CIFAR-100 | | | |
|---|---|---|---|---|
| | 20% *NF* | 30% *NF* | 40% *NF* | 50% *NF* |
| Cross-Entropy | 59.6 | 52.2 | 49.2 | 44.4 |
| Bootstrapping (Reed et al., 2014) | 61.8 | 54.2 | 50.2 | 45.8 |
| Distillation (Li et al., 2017) | 62.7 | 57.3 | 53.7 | 45.7 |
| GLC (Hendrycks et al., 2018) | 68.8 | 65.9 | 62.1 | 57.9 |
| L2RW (Ren et al., 2018) | 67.8 | 63.8 | 59.7 | 55.6 |
| **L2B (Ours)** | **71.8** | **69.5** | **67.3** | **64.5** |

Figure 2: Test accuracy v.s. number of epochs on CIFAR-100 under the noise fraction of 20% and 40%.

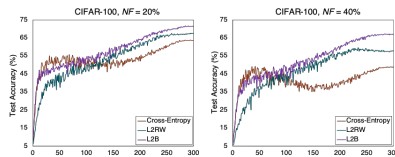

asymmetric noise. Note that we mainly compare with methods which do not use any augmentation techniques, as our L2B does not rely on those. However, we do notice that our method outperforms M-correction which is built on top of mixup augmentation (Zhang et al., 2018), demonstrating the benefits of our joint instance and label reweighting mechanisms for tackling asymmetric noise.

**Alleviate potential overfitting to noisy examples.** We also plot the testing accuracy curve under different noise fractions in Figure 2, which shows that our proposed L2B would help preventing potential overfitting to noisy samples compared with standard training. Meanwhile, compared to simply sample reweighting (L2RW), our L2B introduces pseudo-labels for bootstrapping the learner and is able to converge to a better optimum.

**Generalization to real-world noisy labels.** We test L2B on Clothing 1M (Xiao et al., 2015), a large-scale dataset with real-world noisy labels. To further illustrate the effectiveness of our approach, we compare with: 1) Tanaka et al. (Tanaka et al., 2018), 2) MLNT (Li et al., 2019), 3) MW-Net (Shu et al., 2019), 4) GLC (Hendrycks et al., 2018), and 5) MLC (Zheng et al., 2021) using the same validation and testing splits provided by the benchmark. The results of all competitors are reported from (Zheng et al., 2021). As shown in Table 4, our L2B achieves an average performance of 77.5% accuracy from 3 independent runs with different random seeds, outperforming all competing methods on the Clothing 1M benchmark.

Table 4: Comparison with state-of-the-art methods in test accuracy (%) under real-world noise on Clothing 1M. Our results are reported from 3 independent runs with different random seeds. All other results are reported from (Zheng et al., 2021).

| Method | Tanaka et al. (Tanaka et al., 2018) | MLNT (Li et al., 2019) | MW-Net (Shu et al., 2019) | GLC (Hendrycks et al., 2018) | MLC (Zheng et al., 2021) | **L2B (Ours)** |
|---|---|---|---|---|---|---|
| Accuracy | 72.2 | 73.5 | 73.7 | 73.7 | 75.8 | **77.5 ± 0.2** |

**Generalization to medical image analysis.** Table 5 shows the generalizability of our proposed L2B to medical image analysis (Xue et al., 2019). Compared with 1) L2RW (Ren et al., 2018), which has also been applied to skin lesion classification/segmentation (Xue et al., 2019; Mirikharaji et al., 2019) with noisy supervision; 2) Distillation (Li et al., 2017); 3) Mixup (Zhang et al., 2018); and 4) Bootsrapping (Reed et al., 2014), our method achieves better results under all different noise levels.

**Stability of experimental results.** We repeated the experiments 5 times with different random seeds for network initialization and label noise generation, and report mean ± standard deviation under different experimental settings. As can be seen from Table 6, with different noise fractions and datasets, the standard deviation among the 5 runs are consistently less that 0.5%.

Table 5: Comparison with different methods in test accuracy (%) on ISIC with symmetric noise.

| Method | 20% *NF* | 30% *NF* | 40% *NF* | 50% *NF* |
|---|---|---|---|---|
| Cross-Entropy | 79.4 | 77.5 | 75.3 | 73.7 |
| Bootstrapping (Reed et al., 2014) | 80.8 | 77.7 | 75.7 | 74.8 |
| Distillation (Li et al., 2017) | 80.1 | 78.8 | 76.8 | 74.4 |
| Mixup (Zhang et al., 2018) | 80.2 | 77.9 | 76.8 | 74.9 |
| L2RW (Ren et al., 2018) | 80.1 | 77.7 | 76.3 | 74.1 |
| **L2B (Ours)** | **81.1** | **80.2** | **78.6** | **76.8** |

Table 6: Multiple-runs experiments under the noise fraction of 20% *NF* and 40% *NF*.

| Datasets | 20% *NF* | 40 % *NF* |
|---|---|---|
| CIFAR-10 | 91.99± 0.10 | 89.39± 0.22 |
| CIFAR-100 | 71.78 ± 0.38 | 67.23 ± 0.33 |
| ISIC 2019 | 81.16 ± 0.29 | 78.26 ± 0.45 |

### 4.3 ABLATION STUDY

**On the importance of $\alpha, \beta$.** To understand why our proposed new learning objective can outperform previous meta-learning-based instance reweighting methods, we conduct the following analysis to understand the importance of hyper-parameter $\alpha$ and $\beta$ in our method. Specifically, we set $\alpha = 0$ and $\beta = 0$ respectively to investigate the importance of each loss term in Eq. equation 3. In addition, we also show how the restriction of $\alpha_i + \beta_i = 1$ (Eq. equation 2) would deteriorate our model performance as follows.

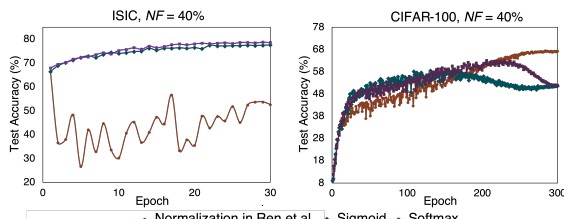

Figure 3: **Comparison among different normalization functions (i.e., Eq. equation 9, Sigmoid function and Softmax function)**. Testing accuracy curv: (a) with different normalization functions under 40% symmetric noise label on the ISIC dataset. (b) with different normalization under 40% symmetric label noise on CIFAR-100.

Table 7: **Ablation study under the noise fraction of 20% and 40%**. L2B ($\alpha, \beta \geq 0$) consistently achieves superior results to L2B ($\alpha + \beta = 1$) under different noise levels on CIFAR-100.

| Method | 20% *NF* | 40% *NF* |
|---|---|---|
| Cross-Entropy | 59.6 | 49.2 |
| $\alpha = 0$ | 55.7 | 47.1 |
| $\beta = 0$ | 63.2 | 57.5 |
| $\alpha + \beta = 1$ | 64.8 | 59.1 |
| $\alpha, \beta \geq 0$ | **71.8** | **67.3** |

- **$\alpha = 0$.** As shown in Table 7, in this case, the performance even decreases compared with the baseline approach. This is due to that when only pseudo-labels are included in the loss computation, the error which occurs in the initial pseudo-label will be reinforced by the network during the following iterations.

- **$\beta = 0$.** From Eq. equation 3, we can see that setting $\beta$ as $0$ is essentially equivalent to the baseline meta-learning-based instance reweighting method L2RW (Ren et al., 2018). In this case, the performance is largely improved compared to the baseline, but still inferior to our method, which jointly optimizes $\alpha$ and $\beta$.

- **$\alpha + \beta = 1$.** We also investigate whether the restriction of $\alpha + \beta = 1$ is required for obtaining optimal weights during the meta-update, as in (Zhang et al., 2020). As shown in Table 7, L2B ($\alpha, \beta \geq 0$) consistently achieves superior results than L2B ($\alpha + \beta = 1$) under different noise levels on CIFAR-100. The reason may be the latter is only reweighting different loss terms, whereas the former not only explores the optimal combination between the two loss terms but also jointly adjusts the contribution of different training samples.

**Parameter normalization.** We note that the normalization of $\alpha$ and $\beta$ is one key component for accelerating the training process. However, we observe that different normalization methods of $\alpha$ and $\beta$ behave quite differently for different datasets. To further investigate this, we apply the following normalization functions to each $\alpha_i$ and $\beta_i$ on ISIC2019, CIFAR-100 and Clothing 1M: 1) Eq. equation 9 as in (Ren et al., 2018), 2) Sigmoid function,

$$\alpha_{t,i} = \frac{1}{1 + e^{-\alpha_{t,i}}}, \ \beta_{t,i} = \frac{1}{1 + e^{-\beta_{t,i}}}, \tag{13}$$

and 3) Softmax function,

$$\alpha_{t,i} = \frac{e^{\alpha_{t,i}/\tau}}{\sum_i^n e^{\alpha_{t,i}/\tau} + e^{\beta_{t,i}/\tau}}, \ \beta_{t,i} = \frac{e^{\beta_{t,i}/\tau}}{\sum_i^n e^{\alpha_{t,i}/\tau} + e^{\beta_{t,i}/\tau}}, \tag{14}$$

where $t$ stands for the training iteration and $\tau$ denotes the temperature parameter for scaling the weight distribution. $\tau$ is set as 10.0 when using the Softmax function for normalization. The comparison among these three different normalization methods is summarized in Figure 3 on ISIC2019

and CIFAR-100 datasets with 40% symmetric noise. We can see that while Eq. equation 9 achieves the best result on CIFAR-100, it yields large training instability on the ISIC2019 dataset. Changing the normalization function to Sigmoid and Softmax can make the training procedure much more stable on the ISIC2019 dataset.

In addition, we notice that Softmax (e.g., using the Softmax function with a temperature scaling of 10.0 instead of Eq. equation 9 to normalize $\alpha$ and $\beta$) slightly outperforms Sigmoid for ISIC2019. Future research should address how to develop a unified normalization function which can be generalizable for different learning tasks.

**Comparison with meta-learning-based label correction.** One recent study (Zheng et al., 2021) has proposed meta label correction to explicitly relabel the noisy examples via a meta-network. Unlike our L2B which conducts implicit relabeling via reweighting different loss components, this method uses a meta-model for explicit label correction. We re-implement the results with the public code by using PreActResNet-18 as the backbone network and the comparison results on CIFAR-10 are show in Table 8. We can see that our method consistently outperforms MLC for noise ratios ranging from 10% to 50% on CIFAR-10.

Table 8: Comparison with MLC (Zheng et al., 2021) on CIFAR-10 with symmetric noise.

| Method | 10% *NF* | 20% *NF* | 30% *NF* | 40% *NF* | 50% *NF* |
|---|---|---|---|---|---|
| MLC | 90.1 | 90.1 | 88.3 | 87.2 | 86.0 |
| **L2B (Ours)** | **93.6** | **92.2** | **90.7** | **89.9** | **88.5** |

**Training cost.** We report our training time on CIFAR-10 compared with the previous meta-learning based instance reweighting methods, i.e., MLNT (Li et al., 2019), L2RW (Ren et al., 2018), in Table 9. The training times are reported based on PreActResNet-18 on a single V100 GPU card. Our method L2B is directly built on top of L2RW, and our method (which applies a joint instance reweighting and label reweighting) incurs almost no extra training cost when using the same architecture and hardware conditions.

Table 9: Training time comparison of MLNT (Li et al., 2019), L2RW (Ren et al., 2018) and L2B (Ours). Our L2B incurs almost no additional training cost when using the same architecture and hardware conditions.

| MLNT (Li et al., 2019) | L2RW (Ren et al., 2018) | **L2B (Ours)** |
|---|---|---|
| 8.6 h | 6.5h | 6.5h |

## 4.4 Discussion & Limitation

L2B is designed for building robust representation models, which is crucial for many real-world applications in healthcare, vision, etc. For example, high intra- and inter-physician variations are well-known in medical diagnostic tasks, which lead to erroneous labels that could derail our learning algorithms. L2B can help rectify such noisy data distributions and prevent potential overfitting during training and thus reduce the risking of medical errors.

This paper specifically focuses on the situation where a small clean validation set is accessible (e.g., Clothing 1M (Xiao et al., 2015)). However, we do note that the extra validation dataset is not a requirement in meta-learning as we can use a subset of pseudo-labeled training data as the validation data (Xu et al., 2021). How to properly select a high-quality pseudo-labeled set and how it affects the algorithm compared to the external validation set should be investigated in the future. Future study should also analyze the role of L2B in conjunction with other meta-learning-based instance reweighting methods such as MW-Net (Shu et al., 2019), and semi-supervised/self-supervised learning methods (Li et al., 2020; 2021), for furthering the performance.

## 5 Conclusion

In this paper, we present L2B, a simple and effective method for training noise-robust models. Concretely, we propose a novel and generic learning objective to enable joint reweighting of instances and labels for combating the label noise in deep representation learning. A meta process is employed to dynamically adjust the per-sample importance weight between real observed labels and pseudo-labels. Our L2B outperforms prior instance reweighting or label reweighting works under both synthetic and real-world noise with almost no extra cost.

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

# A  APPENDIX

## A.1  IMPLEMENTATION DETAILS

For all CIFAR-10 and CIFAR-100 comparison experiments, we used an 18-layer PreActResNet (He et al., 2016) as the baseline network following the setups in (Li et al., 2020), unless otherwise specified. The model was trained using SGD with a momentum of 0.9, a weight decay of 0.0005, and a batch size of 256 for CIFAR-100 and 512 for CIFAR-10. The network was trained from scratch for 300 epochs. We set the learning rate as 0.15 initially with a cosine annealing decay. Following (Li et al., 2020), we set the warm up period as 10 epochs for both CIFAR-10 & CIFAR-100. The optimizer and the learning rate schedule remained the same for both the main and the meta model. Gradient clipping is applied to stabilize training. All experiments were conducted with one V100 GPU, except for the experiments on Clothing 1M which were conducted with one RTX A6000 GPU.

For ISIC2019 experiments, we used ResNet-50 with ImageNet pretrained weights. A batch size of 64 was used for training with an initial learning rate of 0.01. The network was trained for 30 epochs in total with the warmup period as 1 epoch. All other implementation details remained the same as above. For Clothing 1M experiments, we used an ImageNet pre-trained 18-layer ResNet (He et al., 2016) as our baseline. We finetuned the network with a learning rate of 0.005 for 300 epochs. The model was trained using SGD with a momentum of 0.9, a weight decay of 0.0005, and a batch size of 256. Following (Li et al., 2020), to ensure the labels (noisy) were balanced, for each epoch, we sampled 250 mini-batches from the training data.

## A.2  QUALITATIVE RESULTS

We also demonstrate a set of qualitative examples to illustrate how our proposed L2B benefits from the joint instance and label reweighting paradigm. In Figure 4, we can see that when the estimated pseudo label is of high-quality, i.e., the pseudo label is different from the noisy label but equal to the clean label, our model will automatically assign a much higher weight to $\beta$ for corrupted training samples. On the contrary, $\alpha$ can be near zero in this case. This indicates that our L2B algorithm will pay more attention to the pseudo label than the real noisy label when computing the losses. In addition, we also show several cases where the pseudo label is equal to the noisy label, where we can see that $\alpha$ and $\beta$ are almost identical under this circumstance since the two losses are of the same value. Note that the relatively small values of $\alpha$ and $\beta$ are due to that we use a large batch size (i.e., 512) for CIFAR-10 experiments. By normalizing the weights in each training batch (see Eq. equation 9), the value of $\alpha$ and $\beta$ can be on the scale of $10^{-4}$.

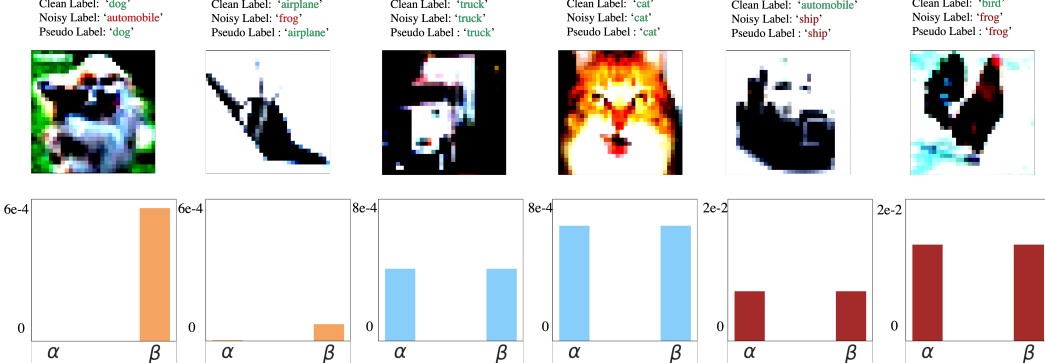

Figure 4: Examples of $\alpha$ and $\beta$ on CIFAR-10 with asymmetric noise fraction of 20%. When the estimated pseudo label is of high-quality, i.e., the pseudo label is different from the noisy label but equal to the clean label, our model will automatically assign a much higher weight to $\beta$ than to $\alpha$ for corrupted training samples. When the pseudo label is equal to the noisy label (i.e., the two loss terms are equal to each other), $\alpha$ and $\beta$ are almost identical.

## B   THEORETICAL ANALYSIS

### B.1   EQUIVALENCE OF THE TWO LEARNING OBJECTIVES

We show that Eq. equation 3 is equivalent with Eq. equation 2 when $\forall i \; \alpha_i + \beta_i = 1$. For convenience, we denote $y_i^{\text{real}}, y_i^{\text{pseudo}}, \mathcal{F}(x_i, \theta)$ using $y_i^r, y_i^p, p_i$ respectively.

$$\alpha_i \mathcal{L}(p_i, y_i^r) + \beta_i \mathcal{L}(p_i, y_i^p) = \sum_{l=1}^{L} \alpha_i y_{i,l}^r \log p_{i,l} \tag{15}$$

$$+ \beta_i y_{i,l}^p \log p_{i,l} = \sum_{l=1}^{L} (\alpha_i y_{i,l}^r + \beta_i y_{i,l}^p) \log p_{i,l} \tag{16}$$

Due to that $\mathcal{L}(\cdot)$ is the cross-entropy loss, we have $\sum_{l=1}^{L} y_{i,l}^r = \sum_{l=1}^{L} y_{i,l}^p = 1$. Then $\sum_{l=1}^{L} \alpha_i y_{i,l}^r + \beta_i y_{i,l}^p = \alpha_i + \beta_i$. So if $\alpha_i + \beta_i = 1$, we have

$$\sum_{l=1}^{L} (\alpha_i y_{i,l}^r + \beta_i y_{i,l}^p) \log p_{i,l} = \mathcal{L}(p_i, \alpha_i y_i^r + \beta_i y_i^p) \tag{17}$$

$$= \mathcal{L}(p_i, (1 - \beta_i) y_i^r + \beta_i y_i^p) \tag{18}$$

### B.2   GRADIENT USED FOR UPDATING $\theta$

We derivative the update rule for $\boldsymbol{\alpha}, \boldsymbol{\beta}$ in Eq. equation 10.

$$\alpha_{t,i} = -\eta \frac{\partial}{\partial \alpha_i} \left( \sum_{j=1}^{m} f_j^v(\hat{\theta}_{t+1}) \right) \Big|_{\alpha_i = 0} \tag{19}$$

$$= -\eta \sum_{j=1}^{m} \nabla f_j^v(\hat{\theta}_{t+1})^T \frac{\partial \hat{\theta}_{t+1}}{\partial \alpha_i} \Big|_{\alpha_i = 0} \tag{20}$$

$$= -\eta \sum_{j=1}^{m} \nabla f_j^v(\hat{\theta}_{t+1})^T \tag{21}$$

$$\frac{\partial (\theta_t - \lambda \nabla (\sum_k \alpha_k \, f_k(\theta) + \beta_k \, g_k(\theta)) \big|_{\theta = \theta_t})}{\partial \alpha_i} \Big|_{\alpha_i = 0} \tag{22}$$

$$= \eta \lambda \sum_{j=1}^{m} \nabla f_j^v(\theta_t)^T \nabla f_i(\theta_t) \tag{23}$$

$$\beta_{t,i} = -\eta \frac{\partial}{\partial \beta_i} \left( \sum_{j=1}^{m} f_j^v(\hat{\theta}_{t+1}) \right) \Big|_{\beta_i = 0} \tag{24}$$

$$= -\eta \sum_{j=1}^{m} \nabla f_j^v(\hat{\theta}_{t+1})^T \frac{\partial \hat{\theta}_{t+1}}{\partial \beta_i} \Big|_{\beta_i = 0} \tag{25}$$

$$= -\eta \sum_{j=1}^{m} \nabla f_j^v(\hat{\theta}_{t+1})^T \tag{26}$$

$$\frac{\partial (\theta_t - \lambda \nabla (\sum_k \alpha_k \, g_k(\theta) + \beta_k \, g_k(\theta)) \big|_{\theta = \theta_t})}{\partial \beta_i} \Big|_{\beta_i = 0} \tag{27}$$

$$= \eta \lambda \sum_{j=1}^{m} \nabla f_j^v(\theta_t)^T \nabla g_i(\theta_t) \tag{28}$$

Then $\theta_{t+1}$ can be calculated by Eq. equation 10 using the updated $\alpha_{t,i}, \beta_{t,i}$.

## B.3 CONVERGENCE

This section provides the proof for covergence (Theorem 1)

**Theorem.** *Suppose that the training loss function $f, g$ have $\sigma$-bounded gradients and the validation loss $f^v$ is Lipschitz smooth with constant L. With a small enough learning rate $\lambda$, the validation loss monotonically decreases for any training batch B, namely,*

$$G(\theta_{t+1}) \leq G(\theta_t), \tag{29}$$

*where $\theta_{t+1}$ is obtained using Eq. equation 10 and $G$ is the validation loss*

$$G(\theta) = \frac{1}{M} \sum_{i=1}^{M} f_i^v(\theta), \tag{30}$$

*Furthermore, Eq. equation 29 holds for all possible training batches only when the gradient of validation loss function becomes 0 at some step t, namely, $G(\theta_{t+1}) = G(\theta_t) \, \forall B \Leftrightarrow \nabla G(\theta_t) = 0$*

*Proof.* At each training step $t$, we pick a mini-batch $B$ from the union of training and validation data with $|B| = n$. From section B we can derivative $\theta_{t+1}$ as follows:

$$\theta_{t+1} = \theta_t - \lambda \sum_{i=1}^{n} (\alpha_{t,i} \nabla f_i(\theta_t) + \beta_{t,i} \nabla g_i(\theta_t)) \tag{31}$$

$$= \theta_t - \eta \lambda^2 M \sum_{i=1}^{n} (\nabla G^T \nabla f_i \nabla f_i + \nabla G^T \nabla g_i \nabla g_i) \tag{32}$$

We omit $\theta_t$ after every function for briefness and set $m$ in section B equals to $M$. Since $G(\theta)$ is Lipschitz-smooth, we have

$$G(\theta_{t+1}) \leq G(\theta_t) + \nabla G^T \Delta\theta + \frac{L}{2}||\Delta\theta||^2. \tag{33}$$

Then we show $\nabla G^T \Delta\theta + \frac{L}{2}||\Delta\theta||^2 \leq 0$ with a small enough $\lambda$. Specifically,

$$\nabla G^T \Delta\theta = -\eta \lambda^2 M \sum_i (\nabla G^T \nabla f_i)^2 + (\nabla G^T \nabla g_i)^2. \tag{34}$$

Then since $f_i, g_i$ have $\sigma$-bounded gradients, we have

$$\frac{L}{2}||\Delta\theta||^2 \leq \frac{L\eta^2 \lambda^4 M^2}{2} \sum_i (\nabla G^T \nabla f_i)^2 ||\nabla f_i||^2 \tag{35}$$

$$+ (\nabla G^T \nabla g_i)^2 ||\nabla g_i||^2 \tag{36}$$

$$\leq \frac{L\eta^2 \lambda^4 M^2 \sigma^2}{2} \sum_i (\nabla G^T \nabla f_i)^2 + (\nabla G^T \nabla g_i)^2 \tag{37}$$

Then if $\lambda^2 < \frac{2}{\eta \sigma^2 ML}$,

$$\nabla G^T \Delta\theta + \frac{L}{2}||\Delta\theta||^2 \leq (\frac{L\eta^2 \lambda^4 M^2 \sigma^2}{2} - \eta \lambda^2 M) \tag{38}$$

$$\sum_i (\nabla G^T \nabla f_i)^2 + (\nabla G^T \nabla g_i)^2 \leq 0. \tag{39}$$

Finally we prove $G(\theta_{t+1}) = G(\theta_t) \, \forall B \Leftrightarrow \nabla G(\theta_t) = 0$: If $\nabla G(\theta_t) = 0$, from section B we have $\alpha_{t,i} = \beta_{t,i} = 0$, then $\theta_{t+1} = \theta_t$ and thus $G(\theta_{t+1}) = G(\theta_t) \, \forall B$. Otherwise, if $\nabla G(\theta_t) \neq 0$, we have

$$0 < ||\nabla G||^2 = \nabla G^T \nabla G = \frac{1}{M} \sum_{i=1}^{M} \nabla G^T \nabla f_i^v, \tag{40}$$

which means there exists a $k$ such that $\nabla G^T \nabla f_k^v > 0$. So for the mini-batch $B_k$ that contains this example, we have

$$G(\theta_{t+1}) - G(\theta_t) \leq \nabla G^T \Delta\theta + \frac{L}{2}||\Delta\theta||^2 \tag{41}$$

$$\leq (\frac{L\eta^2\lambda^4 M^2\sigma^2}{2} - \eta\lambda^2 M) \tag{42}$$

$$\sum_{i\in B}(\nabla G^T \nabla f_i)^2 + (\nabla G^T \nabla g_i)^2 \tag{43}$$

$$\leq (\frac{L\eta^2\lambda^4 M^2\sigma^2}{2} - \eta\lambda^2 M)\nabla G^T \nabla f_k^v \tag{44}$$

$$< 0. \tag{45}$$

