# OpenReview forum: "L2B: Learning to Bootstrap for Combating Label Noise"
_ICLR.cc/2023/Conference — Submitted to ICLR 2023_

### Official Review · Reviewer_LnzH · 2022-10-19

**Confidence:** 2
**Correctness:** 3
**Technical Novelty And Significance:** 3
**Empirical Novelty And Significance:** 2
**Recommendation:** 6

**Clarity, Quality, Novelty And Reproducibility:**

The paper is generally written clearly. The method makes sense to me.



**Strength And Weaknesses:**

strength:
1. the motivation of this paper is valid to me, i.e., simultaneously adjust the per-sample loss weight while implicitly relabeling the training samples
2. the results seem to be effective from the main tables on main benchmarks.
3. the paper is written clearly and is easy to understand.

weakness:
1. A small unbiased and clean validation set is required in this method. Is it a strong assumption made in the current framework?
2. One of the popular and relevant baselines is Dividemix[1], which also involves reweighting mechanism in their approach. Have the authors ever compared their method with this baseline?
3. From fig 4, the weight before the samples is very small. It is interesting to explain why this phenomenon happens. Does that affect the learning procedure? Also, if that is indeed the case, why the authors can claim the data distribution is protected since a lot of samples seem to be ``discarded" as well?

[1] Junnan Li, Richard Socher, and Steven CH Hoi. Dividemix: Learning with noisy labels as semisupervised learning. In ICLR, 2020.


-----post rebuttal

Thank the authors for the detailed response! I would like to raise my score.

**Summary Of The Paper:**

This paper proposes a more generic learnable loss objective which enables a joint reweighting of instances and labels at once, in order to mitigate the effect of noise labels on the model generalization. Specifically, their method dynamically adjusts the per-sample importance weight between the real observed labels and pseudo-labels, where the weights are efficiently determined in a meta process.

**Summary Of The Review:**

The paper meta-learns two weighting parameters for the real label and the pseudo label for combating the label noise problem. There are some vague explanations/experiments in the paper that need to be clarified. Will consider changing my score after I see some of the other reviewers' opinions and the authors' responses.

---

> ### Author Response · Authors · 2022-11-19
> **Response to Reviewer LnzH:**
>
> 1. **Validation:**
>
>    We follow the standard evaluation protocol and use a balanced set of 1000 examples as the validation set, following many prior publications including [7, 10, 11, 23, 25, 40]. We also test our method on a large-scale real-world noisy dataset, Clothing 1M [Xiao et al., CVPR 2015], consisting of 1 million training images collected from online shopping websites with labels generated from surrounding texts.
>
>    It is a stronger but still affordable assumption, considering that the number of training sets is 50K and the validation set is 1K (2% of training data). Also, this assumption follows the standard protocol of meta-learning. Compared with previous manually designed reweighting methods which generally have their own bias, the meta-learning-based algorithm shows superior performance in dealing with class imbalance, noisy labels, and both. Since Ren et al. proposed L2RW in 2018, many follow-ups (including [15,18, 25, 29, 33, 39, 40]) have achieved great results for noisy learning tasks.
>    Therefore, we follow this kind of method and adopt meta-learn to estimate our weighting parameters on validation data.
>
>
> 2. **Comparison to DivideMix:**
>
>    The same concern has been raised by the other two reviewers.
>
>    Our work focus on dealing with noisy label based on meta-learning approaches, which assumes the availability of a small clean dataset and tries to discover useful information from it.
>    It is an orthogonal direction compared with [1] [2] [3] which focuses on the noisy training set only. So to prove the effectiveness of the proposed method, we mainly compare it with meta-learning-based approaches.
>    To address the concern, we further make comparisons with the mentioned methods by the reviewer.
>
>      1. We compare [1] under an asymmetric noise setting. Without a semi-supervised framework and strong augmentation, L2B can achieve superior results.
>      2. We compare [1] [2] [3] on a large-scale real-world benchmark Clothing 1M. Our method can outperform by a notable margin as shown in the following Table.
>      3. We further combine our proposed method with [1] as suggested to show the unique value of our work.
>
>
>
>    > CIFAR 10
>
>    | Methods      | Asym 40% NF |
>    | ------------ | :---------- |
>    | DivideMix[1] | 91.4        |
>    | L2B (Ours)   | **91.8**    |
>
>       On the large-scale real-world benchmark Clothing 1M, our method can outperform [1] [2] [3] [4]  by a notable margin as shown in following Table, which demonstrates the effectiveness.
>
>    | Methods       | Top1           |
>    | ------------- | :------------- |
>    | DivideMix [1] | 74.76          |
>    | RRL [2]       | 74.84          |
>    | PES [3]       | 74.99          |
>    | ELR [4]       | 74.81          |
>    | L2B (Ours)    | **77.5 ± 0.2** |
>
>    Moreover, following the suggestion of reviewer 7ks8, we conduct ablation to combine L2B with DivideMix, results are shown in following Table,
>
>    | Methods      | Epoch | Sym 50% *NF* |
>    | ------------ | :---- | ------------ |
>    | DivideMix[1] | 100   | 93.3         |
>    | + L2B        | 100   | **93.9 (+0.6)**     |
>    | DivideMix[1] | 300   | 94.6                |
>    | + L2B        | 300   | **95.2** **(+0.6)** |
>
>    We can observe that L2B brings consistent improvements (+0.6) to DivideMix, which confirms its effectiveness.
>
>
> 3. **Small weights in Figure 4:**
>
> Thanks for your careful review. These relatively small values of $\alpha$ and $\beta$ are due to the fact that we use a large batch size (i.e., 512) for CIFAR-10 experiments. By normalizing the weights in each training batch (see Eq. 9), the value of $\alpha$ and $\beta$ can be on the scale of $10^{-4}$.  This does not affect the training. On the contrary, large batch size and the use of normalization can largely guarantee the stability of the training. In fact,  $\alpha$ and $\beta$ are directly applied to the CE loss which will sum up all batch so that the scale of  $\alpha$ and $\beta$ won't influence the scale of loss.
>
> We can keep the data distribution since we can still assign the corrupted examples a high weight on the pseudo-label. The sample won't be 'discarded' unless both $\alpha$ and $\beta$ are close to zero. In our experiments, there is almost no such case.
>
>
>
> [1] Li, Junnan, Richard Socher, and Steven CH Hoi. "DivideMix: Learning with Noisy Labels as Semi-supervised Learning." International Conference on Learning Representations. 2020.
>
> [2] Li, Junnan, Caiming Xiong, and Steven CH Hoi. "Learning from noisy data with robust representation learning." In Proceedings of the IEEE/CVF International Conference on Computer Vision, pp. 9485-9494. 2021.
>
> [3] Bai, Yingbin, et al. "Understanding and improving early stopping for learning with noisy labels." Advances in Neural Information Processing Systems 34 (2021): 24392-24403.
>
> [4] Liu, Sheng, et al. "Early-learning regularization prevents memorization of noisy labels." Advances in neural information processing systems 33 (2020): 20331-20342.

---

> > ### Author Response · Authors · 2022-11-24
> > **follow up**
> >
> > Dear Reviewer LnzH,
> >
> > We thank you again for the valuable comments!! Could you please let us know if our rebuttal addresses your concerns? If yes, would you like to raise the score?
> >
> > Thanks
> > Authors

---

> > > ### Author Response · Authors · 2022-12-05
> > > **Thoughts on our response?**
> > >
> > > Dear Reviewer LnzH,
> > >
> > > Thanks again for your comments!!
> > >
> > > Please let us know if our response addresses your concerns. We'd be happy to address any remaining points! If our response has adequately addressed your concerns, we kindly ask you to consider raising the score. Thanks so much!
> > >
> > > Sincerely,
> > >
> > > Authors

---

### Official Review · Reviewer_vvhR · 2022-10-24

**Confidence:** 3
**Correctness:** 3
**Technical Novelty And Significance:** 2
**Empirical Novelty And Significance:** Not applicable
**Recommendation:** 6

**Clarity, Quality, Novelty And Reproducibility:**

The paper is well-written and has nice clarity. The paper proposes a novel technique of weighting losses for combating noisy labels. The significance of the paper is limited since the proposed method's performance is not comparable with the SOTA.

**Strength And Weaknesses:**

Pros

1. The paper is well-written and easy to follow.

2. Theoretical guarantee of the convergence of the proposed method is provided.

Cons

My major concern is that the proposed method is not comparable or compatible with the current SOTA of the methods (of combating noisy labels) which leverage self-supervised learning algorithms and data augmentation [1,2].

The author is trying to rule those works out of the scope of this work. In the related work, [1] is categorized as "approaches which separately handle instance reweighting and label reweighting" In Section 4.2, the author validates themselves by saying "Note that we mainly compare with methods which do not use any augmentation techniques, as our L2B does not rely on those."

However, the methods proposed in [1,2,3] have much better performance than the method in this paper. For example, in the setting of CIFAR-10 50% symmetric noise. The proposed method L2B has an accuracy of 88.5 while the method of [1] has an accuracy of 94.5±0.1

I am concerned about the usefulness of L2B in real practice since it does not help improve the SOTA method.



[1] Junnan Li, Richard Socher, and Steven CH Hoi. Dividemix: Learning with noisy labels as semisupervised learning. In ICLR, 2020.
[2] Junnan Li, Caiming Xiong, and Steven C.H. Hoi. Learning from noisy data with robust representation learning. In Proceedings of the IEEE/CVF International Conference on Computer Vision
(ICCV), pp. 9485–9494, October 2021.
[3] Wang, Zhaoqing, et al. "Exploring set similarity for dense self-supervised representation learning." Proceedings of the IEEE/CVF Conference on Computer Vision and Pattern Recognition. 2022.

**Summary Of The Paper:**

The paper proposes to meta-learn an instance-level weight for real labels and pseudo-labels when learning with noisy labels.

**Summary Of The Review:**

Overall, considering the clarity, novelty, and quality, the current version of the paper is slightly below the acceptance bar.

===

After rebuttal, the author shows that the proposed method can improve the performance of the current SOTA. Thus I raise my score from 5 to 6.

---

> ### Author Response · Authors · 2022-11-19
> **Response to Reviewer vvhR:**
>
> 1. **SOTA**:
>
> The aim of this paper is to investigate a more effective reweighting scheme to tackle the noise label problem, not purely pursue a state-of-the-art performance by combing a set of techniques. To be as fair as possible, we mainly compare with those methods which mostly rely on instance reweighting [28], bootstrapping loss [27], and self-distillation [19] as these methods are directly related to our method.
>
> We did not focus on comparing the performance with mixture model-based reweighting methods or semi-supervised learning methods which use heavy data augmentation. Actually, these methods adopt the model ensemble or heavy data augmentation techniques, which are widely recognized to boost overall performance. In fact, our proposed reweighting method is orthogonal to many existing efforts and by combing them together, we can achieve overall better performance.
>
> 2. **Compared with DivideMix:**
>
> This concern is also mentioned by reviewer 7ks8. Our work focuses on dealing with noisy label based on meta-learning approaches, which assumes the availability of a small clean dataset and tries to discover useful information from it.
> It is an orthogonal direction compared with [1] [2]  which focuses on the noisy training set only. So to prove the effectiveness of the proposed method, we mainly compare it with meta-learning-based approaches.
> To address the concern, we further make comparisons with the mentioned methods by the reviewer.
>
>   1. We compare [1] under an asymmetric nosie setting. Without semi-supervised framework and strong augmentation, L2B can achieve superior results.
>   2. We compare [1] [2]  on a large-scale real-world benchmark Clothing 1M. Our method can outperform by a notable margin as shown in following Table.
>   3. We further combine our proposed method with [1] as suggested to show the unique value of our work.
>
>
>
> > CIFAR 10
>
> | Methods      | Asym 40% NF |
> | ------------ | :---------- |
> | DivideMix[1] | 91.4        |
> | L2B (ours)   | **91.8**    |
>
>    On the large-scale real-world benchmark Clothing 1M, our method can outperform [1] [2]  by a notable margin as shown in following Table, which demonstrates the effectiveness.
>
> | Methods       | Top1           |
> | ------------- | :------------- |
> | DivideMix [1] | 74.76          |
> | RRL [2]       | 74.84          |
> | L2B (ours)    | **77.5 ± 0.2** |
>
> Moreover, following the suggestion of reviewer 7ks8,  we conduct ablation to combine L2B with DivideMix, results are shown in following Table,
>
> | Methods      | Epoch | Sym 50% *NF* |
>    | ------------ | :---- | ------------ |
>    | DivideMix[1] | 100   | 93.3         |
>    | + L2B        | 100   | **93.9 (+0.6)**     |
>
> We can observe that our method brings consistent improvement (+0.6) to DivideMix, which confirms the effectiveness of L2B.
>
> Lastly, we would like to point out that the related work [3] mainly focuses on the pixel-wise semantic segmentation setting (rather than our general image classification setting), which is not directly comparable.
>
>
>
> [1] Li, Junnan, Richard Socher, and Steven CH Hoi. "DivideMix: Learning with Noisy Labels as Semi-supervised Learning." International Conference on Learning Representations. 2020.
>
> [2] Li, Junnan, Caiming Xiong, and Steven CH Hoi. "Learning from noisy data with robust representation learning." In Proceedings of the IEEE/CVF International Conference on Computer Vision, pp. 9485-9494. 2021.
>
> [3] Wang, Zhaoqing, Qiang Li, Guoxin Zhang, Pengfei Wan, Wen Zheng, Nannan Wang, Mingming Gong, and Tongliang Liu. "Exploring set similarity for dense self-supervised representation learning." In Proceedings of the IEEE/CVF Conference on Computer Vision and Pattern Recognition, pp. 16590-16599. 2022.

---

> > ### Author Response · Authors · 2022-11-24
> > **follow up**
> >
> > Dear Reviewer vvhR,
> >
> > We thank you again for the valuable comments!! Could you please let us know if our rebuttal addresses your concerns? If yes, would you like to raise the score?
> >
> > Thanks
> > Authors

---

> > > ### Comment · Reviewer_vvhR · 2022-11-24
> > > **confused about the result**
> > >
> > > I appreciate the author for the effort in providing the additional experiment results. I am still not sure whether the proposed L2B improves the SOTA.
> > >
> > > Mainly, I am confused by the author's results in the third table in the response. I am assuming it is the result of CIFAR-10 since the author did not explain it here. In Dividemix[1]'s original paper, it was reported in Table 1 that, Dividemix has an accuracy of 94.4%-94.6% in the setting of **Sym 50% NF**. While the author now reported that Dividemix has an accuracy of 93.3% and Dividemix + L2B has an accuracy of 93.9%. They are worse than the result in [1]. Did the author use a different setting than [1]? If yes, why not use the same setup as [1]?
> > >
> > > [1] Junnan Li, Richard Socher, and Steven CH Hoi. Dividemix: Learning with noisy labels as semisupervised learning. In ICLR, 2020.

---

> > > > ### Author Response · Authors · 2022-11-24
> > > > **sorry for the confusion**
> > > >
> > > > Dear Reviewer vvhR,
> > > >
> > > > Thanks for your prompt response.
> > > >
> > > > We are sorry for the confusion: in that experiment, we applied a shorter training length (i.e., reduced from 300 epochs to 100 epochs) to get the results quickly. As you suggested, we now applied the default 300 training epochs setup in [1] to re-run experiments, and reported the results as below:
> > > >
> > > >
> > > > | Methods      | Epoch | Sym 50% *NF*        | Sym 80% *NF*    |
> > > > | ------------ | :---- | ------------------- | --------------- |
> > > > | DivideMix[1] | 300   | 94.6                | 93.2            |
> > > > | + L2B        | 300   | **95.2** **(+0.6)** | **94.0 (+0.8)** |
> > > >
> > > >
> > > > We can observe that L2B can consistently and substantially improve the performance of DivideMix [1], which confirms the effectiveness of our method.
> > > >
> > > > We hope this updated result can address your concerns.
> > > >
> > > > [1] Junnan Li, Richard Socher, and Steven CH Hoi. Dividemix: Learning with noisy labels as semisupervised learning. In ICLR, 2020.

---

> > > > > ### Author Response · Authors · 2022-11-28
> > > > > **Could you please reconsider the rating given the updated results?**
> > > > >
> > > > > Dear Reviewer vvhR,
> > > > >
> > > > > Thanks again for your valuable review!
> > > > >
> > > > > Please let us know if our updated result addresses your concerns. If our response has adequately addressed your concerns, we kindly ask you to consider raising the score. Thanks so much!
> > > > >
> > > > > Sincerely,
> > > > >
> > > > > Authors

---

> > > > ### Author Response · Authors · 2022-12-05
> > > > **Feedback on our updated results?**
> > > >
> > > > Dear Reviewer vvhR,
> > > >
> > > > As we have shown in our previous response, DivideMix + L2B can consistently improve results under the exact same setting. Could you please let us know if this result has addressed your concerns?
> > > >
> > > > Thanks so much!!
> > > >
> > > > Sincerely,
> > > > Authors

---

### Official Review · Reviewer_7ks8 · 2022-10-25

**Confidence:** 4
**Correctness:** 4
**Technical Novelty And Significance:** 3
**Empirical Novelty And Significance:** 3
**Recommendation:** 6

**Clarity, Quality, Novelty And Reproducibility:**

The clarity is good.
The novelty is not that attractive. It seems more like using meta-learning to learn loss parameters. Maybe a deeper understanding can be shown to improve the novelty.

**Strength And Weaknesses:**

Strengths:
1)  L2B dynamically adjusts the per-sample importance weight between the given labels and pseudo-labels.
2) The results are somehow higher than some reweighting methods but not other noisy methods.
3) The experiments are done on different datasets.
4) The convergence is analyzed in the paper.

Weakness:
1) should the \alpha_{t,i} and \beta_{t,i} in Eq.10 and algorithm line 12 be \tilde{\alpha_{t,i}} and \tilde{\beta_{t,i}}?
2) Here is a suggestion: an ablation should be made: \mathcal{L}( \mathcal{F}(x_i,\theta), \alpha y_i^{real}+\beta y_i^{psudo})  \alpha and \beta is learned as in your algorithm. This ablation can better analyze whether the re-weighting between the samples is useful or the re-weighting between the label and psuedo-label is useful.
3) comparison of experiments results: maybe previous works [1][2][3] should be compared. Is there any analysis that the results of L2B are not higher compared to [1][2][3]? Since this work is based on the loss design, can L2B be applicable to the current SOTA noisy methods? Can you please show more results based on these works to better illustrate the effectiveness?
4) typo: 'Eq. equation'

[1] Li, Junnan, Richard Socher, and Steven CH Hoi. "DivideMix: Learning with Noisy Labels as Semi-supervised Learning." International Conference on Learning Representations. 2019.
[2] Bai, Yingbin, et al. "Understanding and improving early stopping for learning with noisy labels." Advances in Neural Information Processing Systems 34 (2021): 24392-24403.
[3] Liu, Sheng, et al. "Early-learning regularization prevents memorization of noisy labels." Advances in neural information processing systems 33 (2020): 20331-20342.

**Summary Of The Paper:**

L2B is a learnable loss objective that enables a joint reweighting of instances and labels at once.  L2B dynamically adjusts the per-sample importance weight between the given labels and pseudo-labels in a meta way.

**Summary Of The Review:**

The parameters of the loss terms are adjusted by meta-learning. However, the novelty seems not that good in this work. Further analysis should be made to give a deeper understanding.

---

> ### Author Response · Authors · 2022-11-19
> **Response to Reviewer 7ks8:**
>
> 1. **$\alpha_{t,i}$ and $\beta_{t,i}$ in Eq.10 and algorithm line 12 should be $\tilde{\alpha_{t,i}}$ and $\tilde{\beta_{t,i}}$:**
>
>    Thank you for your meticulous review. We will change the denotations accordingly. Also in (13)(14) we will change to use $\tilde{\alpha_{t,i}}$ and $\tilde{\beta_{t,i}}$ to represent the normalized parameters for sake of consistency.
>
>
>
> 2. **Re-weighting between the samples or labels:**
>
>    The results in Table 7 basically cover this ablation. For the setting $\beta=0$, it only conducts re-weighting between the samples.
>    For the setting $\alpha+\beta=1$, the total weight for each sample is the same $(\alpha_i+\beta_i=1)$, so it can be treated as only re-weighting between the label and pseudo-label.
>    We can see that both re-weighting between the samples and between the label and pseudo-label can bring benefits and combine them together gives the best performance.
>
>
>
> 3. **Comparisons between [1] [2] [3]:**
>
>    Thanks for bringing up these works.  Our work focuses on dealing with noisy label based on meta-learning approaches, which assumes the availability of a small clean dataset and tries to discover useful information from it.
>    It is an orthogonal direction compared with [1] [2] [3] which focuses on the noisy training set only. So to prove the effectiveness of the proposed method, we mainly compare it with meta-learning-based approaches.
>    To address the concern, we further make comparisons with the mentioned methods by the reviewer.
>
>      1. We compare [1] under an asymmetric noise setting. Without a semi-supervised framework and strong augmentation, L2B can achieve superior results.
>      2. We compare [1] [2] [3] on a large-scale real-world benchmark Clothing 1M. Our method can outperform by a notable margin as shown in the following Table.
>      3. We further combine our proposed method with [1] as suggested to show the unique value of our work.
>
>
>
>    > CIFAR 10
>
>    | Methods       | Asym 40% NF |
>    | ------------- | :---------- |
>    | DivideMix [1] | 91.4        |
>    | L2B           | **91.8**    |
>
>       On the large-scale real-world benchmark Clothing 1M, our method can outperform [1] [2] [3] by a notable margin as shown in the following Table, which demonstrates the effectiveness.
>
>    | Methods       | Top1           |
>    | ------------- | :------------- |
>    | DivideMix [1] | 74.76          |
>    | PES [2]       | 74.99          |
>    | ELR [3]       | 74.81          |
>    | Ours          | **77.5 ± 0.2** |
>
>       At last, following your suggestion, we conduct ablation on CIFAR 10 with 50% symmetric noise to see if our method can be applicable to Dividemix :
>
>    | Methods      | Epoch | Sym 50% *NF* |
>    | ------------ | :---- | ------------ |
>    | DivideMix[1] | 100   | 93.3         |
>    | + L2B        | 100   | **93.9 (+0.6)**     |
>    | DivideMix[1] | 300   | 94.6                |
>    | + L2B        | 300   | **95.2** **(+0.6)** |
>
>    We can observe that our method brings consistent improvement (+0.6) to DivideMix, which confirms the effectiveness of L2B.
>
> 4. **Novelty is not that attractive:**
>
>    Our main contribution lies in designing a generalized version of the bootstrapping loss, enabling a joint instance and label reweighting mechanism using the meta-learning framework. We also theoretically prove that our formula (eq. (3)) is in fact a more generalized objective. By reweighting different loss terms, our method conducts an implicit relabeling instead of explicit relabeling in previous works. Compared with L2RW [4] (our meta-learning baseline), our method L2B observes large performance gains, i.e., 3.3% and 7.6% performance improvement in top-1 accuracy for CIFAR-10 and CIFAR-100 classification under 40% symmetric noise.  In a real-world scenario, our method achieves state-of-the-art performance (77.2%) on the Clothing1M benchmark with a notable margin compared with the previous best result (75.8%).
>
> [1] Li, Junnan, Richard Socher, and Steven CH Hoi. "DivideMix: Learning with Noisy Labels as Semi-supervised Learning." International Conference on Learning Representations. 2019.
>
> [2] Bai, Yingbin, et al. "Understanding and improving early stopping for learning with noisy labels." Advances in Neural Information Processing Systems 34 (2021): 24392-24403.
>
> [3] Liu, Sheng, et al. "Early-learning regularization prevents memorization of noisy labels." Advances in neural information processing systems 33 (2020): 20331-20342.
>
> [4] Mengye Ren, Wenyuan Zeng, Bin Yang, and Raquel Urtasun. Learning to reweight examples for robust 360 deep learning. In International Conference on Machine Learning, pages 4334–4343. PMLR, 2018.
>
> [5] Junnan Li, Caiming Xiong, and Steven C.H. Hoi. Learning from noisy data with robust representation learning. In Proceedings of the IEEE/CVF International Conference on Computer Vision (ICCV), pp. 9485–9494, October 2021

---

> > ### Author Response · Authors · 2022-11-24
> > **follow up**
> >
> > Dear Reviewer 7ks8,
> >
> > We thank you again for the valuable comments!! Could you please let us know if our rebuttal addresses your concerns? If yes, would you like to raise the score?
> >
> > Thanks Authors

---

> > ### Comment · Reviewer_7ks8 · 2022-11-27
> > **Respose**
> >
> > Thanks to the authors for their detailed responses, which addressed my concerns and I am therefore inclined to accept the paper. It is best to open source the code so that we can confirm whether the experimental results can be reproduced.

---

> > > ### Author Response · Authors · 2022-11-28
> > > **Thanks for raising your score!**
> > >
> > > Thank you so much for raising your score. We are glad that your concerns get addressed!
> > >
> > > Yes, we will start to clean and release the code as soon as possible.
> > >
> > > Thanks again for your support!

---

### Author Response · Authors · 2022-11-19
**General Response:**

We thank the reviewers for their valuable comments. We appreciate that our paper is recognized to be with valid motivation (LnzH)  novel technique (vvHR), clear method (LnzH),  theoretical supported (vvHR, 7ks8), and effective results (LnzH). Moreover, all the reviewers recognize that this paper is well-organized, has nice clarity and is well-written.  We hope our feedback can address all concerns.

---

### Decision · Program_Chairs · 2023-01-20

**Decision:**

Reject

**Justification For Why Not Higher Score:**

This paper initially received the scores of 5,5,5 ('marginally below the acceptance threshold' x3). One of the reviewers’ main concerns is regarding the novelty of the method since tuning loss parameters (weights) by mete-learning is a relatively well-studied approach, although the reviewers also acknowledge the contribution of the method in the context of the bootstrapping loss. Another concern is about the framing of the study and experimental evaluation. In particular, it was not clear whether the method is comparable or compatible with other noisy learning (i.e., other than meta-learning based ones). The AC also agrees that those concerns are fair.
During the discussion period, the authors responded with additional results which mitigated some of the concerns, and all reviewer decided to raise the score to 6 (marginally above acceptance threshold).  As a result, the final scores are 6,6,6, which are all slightly positive yet the paper still lacks a strong support. Given the high standard of ICLR and the reviewers' opinions, the paper is still on the borderline but could be accepted depending on the availability of the last rooms.


**Justification For Why Not Lower Score:**

N/A

**Metareview: Summary, Strengths And Weaknesses:**

Summary:
This paper proposes a method to learn with noisy labels by jointly reweighting instances and labels at once. To this end, the proposed method (L2B) meta-learns two weighting parameters in the loss for the real label and the pseudo label. In the experiments, L2B is evaluated on standard datasets and shown to outperform some existing methods.

Strengths:
1. The proposed method, L2B, is well motivated and provides a more general form of the bootstrapping loss.
2. Theoretical guarantee of the convergence of the proposed method is provided.
3. The paper is generally written clearly.

Weaknesses:
1. Novelty of the method seems to be in a somewhat narrow context and not ground-breaking.
2. Positioning of the study as well as comparison/compatibility with other noisy label method can be better investigated.